# Sensing at the Nanoscale Using Nitrogen-Vacancy Centers in Diamond: A Model for a Quantum Pressure Sensor

**DOI:** 10.3390/nano14080675

**Published:** 2024-04-12

**Authors:** Hari P. Paudel, Gary R. Lander, Scott E. Crawford, Yuhua Duan

**Affiliations:** 1National Energy Technology Laboratory, United States Department of Energy, Pittsburgh, PA 15236, USA; gary.lander@netl.doe.gov (G.R.L.); scott.crawford@netl.doe.gov (S.E.C.); 2NETL Support Contractor, 626 Cochrans Mill Road, Pittsburgh, PA 15236, USA; 3NETL Support Contractor, 3610 Collins Ferry Road, Morgantown, WV 26505, USA

**Keywords:** quantum sensing, stress sensor, nitrogen-vacancy center in nanodiamond, density functional theory, quantum manometer

## Abstract

The sensing of stress under harsh environmental conditions with high resolution has critical importance for a range of applications including earth’s subsurface scanning, geological CO_2_ storage monitoring, and mineral and resource recovery. Using a first-principles density functional theory (DFT) approach combined with the theoretical modelling of the low-energy Hamiltonian, here, we investigate a novel approach to detect unprecedented levels of pressure by taking advantage of the solid-state electronic spin of nitrogen-vacancy (NV) centers in diamond. We computationally explore the effect of strain on the defect band edges and band gaps by varying the lattice parameters of a diamond supercell hosting a single NV center. A low-energy Hamiltonian is developed that includes the effect of stress on the energy level of a ±1 spin manifold at the ground state. By quantifying the energy level shift and split, we predict pressure sensing of up to 0.3 MPa/$\sqrt{\mathrm{Hz}}$ using the experimentally measured spin dephasing time. We show the superiority of the quantum sensing approach over traditional optical sensing techniques by discussing our results from DFT and theoretical modelling for the frequency shift per unit pressure. Importantly, we propose a quantum manometer that could be useful to measure earth’s subsurface vibrations as well as for pressure detection and monitoring in high-temperature superconductivity studies and in material sciences. Our results open avenues for the development of a sensing technology with high sensitivity and resolution under extreme pressure limits that potentially has a wider applicability than the existing pressure sensing technologies.

## 1. Introduction

Color centers in solid-state physics have attracted significant attention due to the possibility of using them for emerging quantum technologies. These color centers in certain solid materials show promise for applications ranging from magnetic field sensing with unprecedented sensitivity levels using magnetometry to cellular biomarker-based biology investigations, to quantum communication and quantum computing using optically addressable solid-state qubits [1,2,3,4]. Moreover, emerging applications in the energy sector such as the expansion of smart grids/meters, driverless vehicles, and nuclear reactors and the discovery of new oil and gas deposits are creating new opportunities for quantum sensing [5,6]. The continued maturation of quantum sensing technologies offers exciting opportunities for quantum-enhanced measurements that may provide significant improvements in sensitivity beyond the classical limits. Variables such as temperature, pH, electromagnetic fields, and pressure must be measured with high precision, often in harsh conditions (e.g., highly corrosive environments due to high temperature, pressure, humidity, and radioactivity). These sensors are deployed in infrastructures such as transformers [7], pipelines [8,9,10], mines [8], nuclear power plants [11,12] and in other areas [13,14] to ensure safe operating conditions and uninterrupted, optimized service.

Nitrogen-vacancy (NV) centers in diamond are among the most promising color centers in the solid state that hold rich physics and exhibit potential for sensing [2,6,15,16,17,18,19,20,21] and communication [22,23] applications. In addition, the negatively charged NV centers persistently maintain their desirable electronic features at room temperature, unlike many other solid-state systems for which elevated temperatures greatly reduce the signal-to-noise ratio [24,25,26]. Sensors based on NV centers are also proven to surpass the stress sensing limits that are achievable by classical sensing devices [27,28,29,30,31,32,33]. The NV center primarily appears in the negative (NV^−^) and neutral (NV°) charge states within the diamond lattice. Due to the tunability of the qubits defined within the NV centers, these defects are promising candidates for the sensing of chemical analytes. Encapsulation of NV center-containing nanodiamonds with materials such as metal–organic frameworks (MOFs) can provide additional control over the dispersion of the particles and may selectively bring the target analytes to the nanoparticle surfaces for enhanced performance in sensing applications [24,34,35].

A NV center is a six-electron system. Within the NV center, three electrons of the neighboring C atom are linked with the N atom forming covalent bonds, while the remaining two electrons are provided from a lone pair. The sixth electron that results in the formation of a negative charge state is assumed to have been accepted from a nearby donor. Overall there are four ${sp}^{3}$ orbitals, which are labeled $a=\left\{a_{1}^{\prime },a_{1}\right\}$ and $e=\left\{e_{x},e_{y}\right\}$ [36]. The NV centers’ defect bands that appear within the diamond band gap are composed of a triplet (S = 1) ground state where the $m_{s}=\pm 1$ states are degenerate and are separated from the $m_{s}=0$ state by a zero-field splitting energy parameter (D). The spin–spin interaction splits the ground state spin sublevels by 2.88 GHz into a spin singlet S_z_, where z corresponds to the NV center symmetry axis, and a spin doublet S_x_, S_y_ [33]. The excited state ^3^E is also a spin triplet and orbital doublet, which are associated with broadband photoluminescence emission with a zero-phonon line (ZPL) centered around 637 nm. However, the ^3^E level has more complex structures than the ground level ^3^A. At elevated temperatures, the six-level ^3^E fine structure is observed via high-resolution optical spectroscopy. While conducting optically detected magnetic resonance (ODMR) at temperatures above 150 K, the population transfer in the ^3^E orbitals is extremely fast. The ODMR for this level, therefore, resembles that of ^3^A. 

Under 532 nm excitation, carriers are pumped to the excited states where spin is conserved due to the conservation of the total angular momentum. The excited carriers relax to the ground state either directly by emitting photons of 637 nm, originating from the ZPL, or by nonradiative intersystem crossings (ISCs) through long-lived metastable states. The probability of carrier relaxation from the metastable state to the $\left.|A,0\right\rangle$ state is higher compared to that of the decay to the $\left.|A,\pm 1\right\rangle$ ground state, as shown in Figure 1. Under continuous optical excitation, these spin-conserving transitions yield higher populations in the $\left.|A,0\right\rangle$ state and initialize a spin-polarized ground state. This allows for spin-dependent fluorescence measurements using the ODMR technique [19,33,37,38]. The dark ISCs are not yet fully understood; however, it is believed these are the result of cumulative effects from the spin–orbit interaction of the lowest triplet (^3^A, ^3^E) and singlet (^1^A, ^1^E) levels as well as of electron–phonon interactions [33,39,40].

Diamonds that host NV^−^ centers have a high Young’s modulus. This provides excellent mechanical strength under applied stress and hence results in a high stress amplitude relative to that achievable with other widely used materials such as steel, silicon, and ceramics in pressurized environments. In environments where high pressure is essential, NV centers in diamond can be implemented to probe temperature, pressure, and even the strength of an electromagnetic field. In 2014, Doherty et. al. revealed for the first time the possibility of implementing the NV^−^ center for high-pressure measurements up to 60 GPa by probing optical and spin resonances [41]. The zero-field splitting between the $m_{s}=0$ and $m_{s}=\pm 1$ spin sublevels with a value of D~2.88 GHz  mainly arise due to the first-order spin–spin interaction, whereas the second-order spin–spin interaction provides a negligible contribution to the splitting parameter [22,42]. The spin quantization axis is determined by the spin density within the triagonal axes in the ground ^3^A level. Depending on the direction of the applied strain to the NV center symmetry, the degeneracy of the spin sublevel is lifted. The strain along the center symmetry axis results in shifts in the A spin sublevels, and the strain along the traverse direction lifts the degeneracy of the E spin sublevels. In previous studies [1,43,44,45,46,47,48], theoretical models of the NV centers were able to describe experimental observations in nanoscale magnetic field, pressure, and temperature sensing. Ivady et al. [1] used an ab initio method to study the pressure dependence of the zero-field splitting due to the change in the distance between the spins as the applied compression changed. 

Multiple studies were experimentally [48,49,50,51] and theoretically [46,47,52] conducted to capture the change in the zero-phonon line and spin levels. While a plethora of these studies provide insights on the NV centers’ behavior under stressed and harsh environmental conditions, a combined analytical and computational modelling approach could directly help in designing ultrasensitive quantum sensing devices useful in stressed environments, considering the properties of the electronic spin sublevels. In addition, the strength and sensitivity level of a quantum sensor compared to that of a classical sensor still needs to be quantified. Here, we employed a density functional theory (DFT) approach to study the effect of stress on the band edges and band gaps of the NV-defective diamond lattice. Stress in a supercell of single-NV-defective bulk diamond was introduced by varying the strain along the longitudinal (c) and the transverse (a and b) lattice directions for different dimensions of the supercell. Band edge shifts up to several terahertz (THz) were obtained. To capture the shift and splitting in the spin manifold due to stress, we introduced a low-energy Hamiltonian and derived its effective form. Using the diamond’s low-energy Hamiltonian for the NV center, level shift and splitting per unit stress were calculated, and their applicability to stress sensitivity using a previously measured spin dephasing time reported by our group is discussed. By combining the DFT and Hamiltonian modelling results, we also discuss the superiority of quantum sensors for pressure monitoring compared to traditional optical sensing devices based on band gap and band edge tuning. 

## 2. Computational Method

The first-principles density functional theory (DFT) approach is a widely used technique based on the single-particle Kohn–Sham (KS) equation [2], which employs the plane-wave basis sets. This approach is particularly successful in determining the ground-sate properties of solids. In KS DFT, the total energy E is a function of the electron charge density n(r) of the interacting system. The charge density of an interacting system can be expressed in terms of the KS orbital of a non-interacting system as $n\left(r\right)=\sum_{i}c_{i}{{|\varphi }_{i}|}^{2}$, where $\varphi_{i}$ is the KS wave function, and $c_{i}$ is the occupation number of the KS orbital. $\varphi_{i}$ is the eigenfunction of the single-particle Hamiltonian $h_{i}\left(r_{i}\right).$ The total effective Hamiltonian $H_{eff}$ is the sum of $h_{i}\left(r_{i}\right)$ over the number of electrons, $H_{eff}=\sum_{i=1}^{N_{el}}h_{i}\left(r_{i}\right)$, and is expressed by
(1)$$H_{eff}=\sum_{i=1}^{N_{el}}\left\{-\frac{1}{2}\nabla_{i}^{2}+V_{eff}(r_{i})\right\}$$

Due to a lack of accurate exchange–correlation potential, in most cases, the calculated results for excited states using the KS DFT approach are in practice only a rough approximation of the experimentally observed results. While simulating in three-dimensional periodic systems, the KS orbitals are usually expanded in terms of plane-wave basis sets. Numerically, the use of plane-wave basis sets allows for a fast convergence in the ground-state energy. However, due to the requirement of extremely short wavelengths to account for the core electronic orbital, such as 1s in C in our calculation, a large value of kinetic energy (or a large number of basis sets) is necessary, which results in a high computational cost in the numerical calculations. In the projector augmented wave method [53], Bloch developed a soft potential that can be used to reconstruct an all-electron effect, i.e., the effect of core electrons plus valence electrons [54]. Exchange–correlation potentials in the DFT approach are approximated at different theoretical levels. In the local-density approximation (LDA), the exchange–correlation potential is derived for homogenous gas systems [55]. An extension of the LDA is a generalized gradient approximation [56], which considers the variation in the electronic potential by implementing its gradient. In the generalized gradient approximation (GGA), the Perdew–Burke–Ernzerhof (PBE) functional is one of the widely accepted functionals for calculations while considering a tradeoff between the accuracy and the computational time [57]. A major drawback of these methods is that they all suffer from a self-interaction error that usually results in an underestimation of the electronic band gap of solids. 

A cubic structure of diamond with a unit cell experimental lattice parameter a=0.356 nm was considered [58]. In our calculation, the first-principles DFT approach was implemented in the Vienna ab initio simulation package (VASP) [59,60]. The projector augmented wave [53] pseudo-potentials and the Perdew–Burke–Ernzerhof (PBE) exchange–correlation functional were chosen in order to calculate the electronic density of states and bandstructures [54,57]. Plane-wave basis sets with a cutoff energy of 520 eV were used. A pure diamond unit cell was optimized with a Monkhorst–Pack grid of 9 × 9 × 9, and energy convergence of 10^−6^ eV per unit cell was achieved. To calculate the bulk electronic density of states (DOS) and bandstructures for NV-defective diamond, a 3 × 3 × 3 supercell was used along with a Monkhorst–Pack grid of 1 × 1 × 1. Varying uniaxial and isotropic stresses were applied, and the corresponding bandstructures were computationally monitored. A q=−1 charge was introduced to ensure a negative charge state of the NV center to resemble the experimental conditions; this step produced the required defect bands. Bader charge analysis was performed, and the distribution of charge around the NV centers was calculated. In solid and molecular systems, the charge density between atoms has a minimum value due to a vanishing overlap in their wavefunctions near a neutral region that separates the two atoms. The charge density distribution was calculated using $∆\rho =\rho \left({NV}^{-}\right)-\rho (NVnetral)$, where $\rho \left({NV}^{-}\right)$ is the charged density for a supercell with an ${NV}^{-}$ center, and ρ(NVnetral) is the charge density for a neutral supercell. A varying stress was introduced by changing the lattice parameters along the longitudinal and transverse directions of a supercell. 

## 3. Results and Discussions

### 3.1. Density of States and Bandstructures of NV-Defective Diamond

The NV center in the diamond unit cell had $C_{3v}$ symmetry. Under zero applied stress, the triagonal axis of the NV center in $C_{3v}$ symmetry was aligned to the [111] direction, as shown in Figure 1a. Here, we denote the NV^−^ center as the NV center for convenience. The Jahn–Teller effect reduces the neutral NV center (NV°) into the $C_{1h}$ symmetry relaxed state [61,62]. Depending upon the direction of the applied stress, the $C_{3v}$ symmetry either remains intact or is broken. Under applied isotropic stress $\sigma_{xyz}$, the $C_{3v}$ symmetry was preserved. The NV center oriented along the [111] direction preserved the $C_{3v}$ symmetry when the stress was applied along the [111] direction, whereas the $C_{3v}$ symmetry was broken for other orientations, namely, $\left[\bar{1}1\bar{1}\right]$, $\left[\bar{1}1\bar{1}\right]$, and $\left[\bar{1}\bar{1}1\right]$, of the NV center. 

With one NV center in our supercell, the total density of the defects was ${8.163\times 10}^{20}/{cm}^{3}$. When a N and C vacancies were introduced with an additional total charge q=−1 (usually captured from the lattice site in the sample) and spin S=1, this additional charge was aligned along the NV axis, providing the dipole direction. In that case, the net charge density was equal to the density of the NV centers, which was ${8.163\times 10}^{20}/{cm}^{3}$. The charge was distributed around the vacancy, forming dangling bonds with the C atoms surrounding the NV center. In this alignment, the dipole was directed to the N atom, with about 80% of the additional −1 charge localized around the surrounding C atoms. This caused the N states in the NV center to be further shifted toward the valence band by about 0.1 eV. N impurity sp states arose near the valence band, whereas some C states provided significant contribution to the states that appeared near the Fermi level, as shown in Figure 1. A N impurity in pure diamond usually provides an additional charge and therefore acts as an electron donor. These N impurity states appear near the conduction band edge of bulk diamond. When a C vacancy was introduced adjacent to the N impurity, the degenerate excited triplet ^3^E bands in the ${NV}^{-}$ were found to be 60 meV above and the degenerate ground triplet ^3^A 1.48 eV below the Fermi level at the Γ point. The contribution of the impurity to the total DOS near the Fermi level was small, as seen in Figure 1, and the bands near the Fermi level were partially occupied. These bands are essentially the color center’s energy level that make green light excitation (~λ=532 nm) possible during experiments [33,52]. The underestimation of the band gap at the PAW-PBE level is inherent in the DFT calculation.

A systematic change in the lattice parameter along specific directions by allowing ions to relax inside the supercell structures provides an estimate of the independent components of the stress tensor $\sigma_{ij}$. To gain insights into the splitting of the bands due to the exertion of tensile strain, here, we changed the lattice parameters along the longitudinal (c−axis) and traverse (ab plane) directions. Figure 2 shows the bulk diamond bandstructures with N, NV^0^ and NV^−1^ in a 3 × 3 × 3 supercell. We changed c up to ±2%. Due to the finite size of the supercell, $\sigma_{ij}=-\delta E/\delta \eta_{ji}$ was needed to equilibrate the geometry considered in the calculation. At the equilibrium 3×3×3 supercell structure, the magnitude of the diagonal elements of $\sigma_{ij}$ (i,j=x,y,z) was 2.7 GPa, whereas that of the other three independent off-diagonal elements $\sigma_{xy}$, $\sigma_{yx}$, and $\sigma_{zx}$ was 0.3 GPa. The positive sign here indicates compressive strain (and the negative sign represents tensile strain). The values of the applied stress for nonequilibrium lattice parameters could be estimated by subtracting the above components of the stress tensors. 

A tensile strain with +2% change in c exerted a stress of $\sigma_{zz}=-15.194$ GPa, while the magnitude of the other elements was less than +1 GPa. Under compressive strain with −2% change in c, we found $\sigma_{zz}=19.85$ GPa. while the magnitude of the other elements was less than −1 GPa. We also changed the traverse lattice parameters up to ±2%. In this case, $\sigma_{xx}$ and $\sigma_{yy}$ were found to be −16.49 GPa, and $\sigma_{zz}$ was −13.37 GPa. When the supercell structure experienced tensile strain under −2% changes in the transverse lattice parameters, $\sigma_{xx}$ and $\sigma_{yy}$ were found to be 26.61 GPa, and $\sigma_{zz}$ was 5.65 GPa. All other components were less than 2 GPa. 

Figure 3 shows the splitting of bands due to the stress that was applied by changing the transverse lattice parameters (a,b). The stress was found to split the excited ^3^E level (${∆E}_{ex}$), whereas the degenerate ^3^A level was unaltered. ${∆E}_{ex}$ was found to be 40 meV under the compressive and tensile strain introduced by a 2% change in the transverse lattice parameters. ${∆E}_{ex}$ was found to be 60 meV under tensile strain and 30 meV under compressive strain with a 2% change in the c parameter. Figure 4 shows the splitting ${∆E}_{ex}$ of the excited ^3^E level as a function of change in the longitudinal and transverse lattice parameters. The slight asymmetric nature of the splitting shows that the NV center’s response was different under compressive and tensile strain. It is to be noted that stress in the lattice acted with respect to the crystallographic coordinate system. The NV center dipole was not aligned to the direction of stress and therefore affected the polarization of the dipole. Table 1 summarizes the splitting ${∆E}_{ex}$ and changes in the band gap under variations in the lattice parameters.

### 3.2. Theoretical Model of the NV Center under Stress

The DFT approach was useful to gain insights on the bandstructures and energy level splitting under compressive and tensile strain. However, a detailed picture of the splitting of the spin manifold at the ground level required an additional analytical approach based on a complete Hamiltonian that would capture zero-field splitting as well as perturbations in the spin states. In the literature, the spin–strain and spin–stress interaction are described in reference to the crystal axes (XYZ) or the NV center’s local symmetry axes (xyz) [27,46,49]. In order to formulate a theoretical model, we followed the coordinate system and description provided by Udvarhelyi et al. [46]. 

Consider a cubic frame of reference for a NV center in nanodiamond with a N atom at the origin and a vacancy at (a/4, a/4, a/4), where a=3.567 Å is the unit cell parameter. The axis systems are shown in Figure 5. Here, $\left\{x,y,z\right\}$ indicates the NV center’s threefold rotation symmetry axes, and {X,Y,Z} the crystal’s cubic frame of reference. In the {x,y,z} frame, three orthonormal vectors are defined as $e_{z}=(1,\;1,\;1)$/$\sqrt{3}$, $e_{y}=\left(1,-1,0\right)$/$\sqrt{2}$, and $e_{x}=(-1,-1,\;2)$/$\sqrt{6}$. In this frame of representation, the $C_{3}$ symmetry of the NV center lies on the xz plane. It is to be noted that the chosen orientation is one of the four possible NV axis orientations. The transformation between crystal symmetry and NV symmetry defined by the above three orthonormal vectors could be obtained by using the transformation vector $T_{NV}=R_{\left[001\right]}(-3\pi /4)R_{\left[\bar{1}10\right]}\left(-\alpha_{NV}\right)$, where $R_{n}(\theta )$ is a rotation matrix, and $\alpha_{NV}=$ arccos $\left(1/\sqrt{3}\right)$ [27]. Defining the rotation vector $K_{e_{z}}$ for $e_{z}\in \{111,\;\bar{1}\bar{1}1,\;\bar{1}1\bar{1},1\bar{1}\bar{1}\}$, one can generate a transformation among four unique orientations of the NVs via operating $K_{k}$ by $T_{NV}$. For example, the transformation between two NVs with their dipoles oriented along [111] and $\left[\bar{1}\bar{1}1\right]$ can be made by implementing the operation $T_{NV}K_{\left(\bar{1}\;\bar{1}\;1\right)}$. In this case, $K_{\left(\bar{1}\;\bar{1}\;1\right)}=R_{\left[001\right]}(\pi )$. The details of this transformation were presented by Barfuss et al. [27]. Using the NV orientation along the [111] direction, we developed the formalism for coupling the stress to the spins at the NV center’s ground level. The change in stress per spin energy was deduced to quantify the potential sensitivity. 

The negatively charged NV center had spin S=1. The spin sublevels $\left.|0\right\rangle ,\;\left.|-1\right\rangle$ and $\left.|+1\right\rangle$ are eigenstates of the spin operator $S_{z}$ along the z-axis. The spin sublevels $\left.|\pm 1\right\rangle$ are degenerate and are separated by the zero-field splitting parameter $D_{0}=2.87$ GHz from $\left.|0\right\rangle .$ The symmetry-breaking magnetic field $B=(B_{x},B_{y},B_{z})$ induces a Zeeman shift, splitting the $\left.|-1\right\rangle$ and $\left.|+1\right\rangle$ states by $\pm \gamma_{e}B_{z}$, where $\gamma_{e}=$2.8 MHz/G is the electron’s gyromagnetic ratio. In the presence of a magnetic field, neglecting the zero-phonon line, the ground state of the negative NV center is described by the Hamiltonian
(2)$$H_{o}=D_{0}S_{z}^{2}+\gamma_{e}B\cdot S$$
where $S=(S_{x},\;S_{y},\;S_{z})$ is the spin matrix. Only the z component of the field splits the ground state, whereas the transverse components of the field have a negligible effect due to a weak coupling of the field with the $S_{x}$ and $S_{y}$ spin components. 

In order to describe the spin-mechanical coupling, the electronic levels were perturbed by applying a uniaxial stress V = $\sum_{i,j}A_{ij}\sigma_{ij}$, where $A_{ij}$ indicates electronic operators, and $\sigma_{ij}$ indicates second-rank symmetric stress tensors [49,64]. If uniaxial pressure P is applied to the crystal along an arbitrary direction, the elements of the stress tensors become $\sigma_{ij}=P$Cos$\left(p,i\right)\;*$ Cos $\left(p,j\right)$, where p is the direction of the applied pressure. The perturbation V can also be written in terms of a second-rank strain tensor $V=\sum_{i,j}B_{kl}\varepsilon_{kl}$, where $B_{kl}$ indicates the electronic operators. $A_{ij}$ and $B_{kl}$ are related by the elastic constant $c_{ijkl}$, as $B_{kl}=\sum_{i,j}A_{ij}c_{ijkl}$. Due to the symmetry of the stress tensor, there are only six $A_{ij}$, and the elastic constants can be reduced to 6×6 tensors. For cubic crystals such as diamond, there are only three independent components of elastic constants, which greatly simplifies $c_{ijkl}$. The spin states $\left.|0\right\rangle$ and $\left.|\pm 1\right\rangle$ at the ground ^3^A_2_ level of the NV center transform, respectively, as A and E irreducible representations in triagonal symmetry. The theoretical description of tetragonal and triagonal symmetry is equivalent [64]. $\sigma_{ij}$ has two components, ${\{A}_{1},$ $A_{1}^{\prime }$} that transforms as A and another two components, $E_{x}$ and $E_{y}$, that transform as E and also have two components each, ${\{E}_{x},$ $E_{x}^{\prime }\}$ and ${\{E}_{y},\;E_{y}^{\prime }\}$. To be consistent with the notations in the literature, we define $M_{z}=A_{1}+A_{1}^{\prime }$, $M_{y}=E_{y}+E_{y}^{\prime }$, and $M_{x}=E_{x}+E_{x}^{\prime }$. 

The effect of the applied stress at the NV center can be estimated by calculating the distortion on the unpaired spin density that leads to a change in the spin–spin interaction. The states $\left.|0\right\rangle ,\;\left.|\pm 1\right\rangle$ are the eigenfunctions of the spin operator S that also transform as A and E. Following Reference [27], the most general spin–stress coupling Hamiltonian under applied stress in terms of spin components can be written as $H_{\sigma }=H_{\sigma 0}+H_{\sigma 1}+H_{\sigma 2}$, where
(3)$$\frac{H_{\sigma 0}}{h}=M_{z}S_{z}^{2}$$
(4)$$\frac{H_{\sigma 1}}{h}=N_{x}\left\{S_{x},S_{z}\right\}+N_{y}\left\{S_{y},S_{z}\right\}$$
(5)$$\frac{H_{\sigma 2}}{h}=M_{x}\left(S_{y}^{2}-S_{x}^{2}\right)+M_{y}\left\{S_{x},S_{y}\right\}$$
where $\left\{S_{i},S_{j}\right\}=\left(S_{i}S_{j}+S_{j}S_{i}\right)$ is the anticommutator for the spin operators. $M_{i}$ are the spin–stress coupling parameters written in the NV frame of reference, as defined above. The quantity $N_{i}$, not defined above, indicates the small perturbations that arise under the symmetry-breaking field. In the absence of such a field, $N_{i}$ provides a negligible contribution. The strength of the symmetry-breaking field, e.g., a magnetic field, is minimized to less than 1 mT to reduce the induced-field contribution in $H_{\sigma }$ [65]. In Reference [46], a detailed description of the term $H_{\sigma 1}$ that contains $N_{i}$ was provided. The indices 0, 1, and 2 represent changes in $m_{s}$. $H_{\sigma 1}$ has a nonzero dipole transition element between $\left.|0\right\rangle$ and $\left.|\pm 1\right\rangle$ and can couple to a homogeneous electric field. $H_{\sigma 2}$ couples the spin sublevels $\left.|+1\right\rangle$ and $\left.|-1\right\rangle$. It is to be noted that the term $H_{\sigma 0}$ preserves the symmetry of the NV center and brings a constant shift on the $\left.|\pm 1\right\rangle$ level in reference to the $\left.|0\right\rangle$ level. The spin–stress coupling parameters $M_{i}$ and $N_{i}$ in the NV frame of reference can be written in terms of stress tensors expressed with respect to the crystal axes as
(6)$$M_{x}=b\left(2\sigma_{ZZ}-\sigma_{XX}-\sigma_{YY}\right)+c\left(2\sigma_{XY}-\sigma_{YZ}-\sigma_{XZ}\right)$$
(7)$$M_{y}=\sqrt{3}b\left(2\sigma_{XX}-\sigma_{YY}\right)+\sqrt{3}e\left(2\sigma_{YZ}-\sigma_{XZ}\right)$$
(8)$$M_{z}=a_{1}\left(\sigma_{XX}+\sigma_{YY}+\sigma_{ZZ}\right)+2a_{2}\left(\sigma_{XY}+\sigma_{YZ}+\sigma_{XZ}\right)$$
(9)$$N_{x}=d\left(2\sigma_{ZZ}-\sigma_{XX}-\sigma_{YY}\right)+e\left(2\sigma_{XY}-\sigma_{YZ}-\sigma_{XZ}\right)$$
(10)$$N_{y}=\sqrt{3}d\left(2\sigma_{ZZ}-\sigma_{XX}\right)+\sqrt{3}e\left(2\sigma_{YZ}-\sigma_{XZ}\right)$$

The spin–stress interactions described by Equations (5) through (9) contain the stress susceptibility parameters $a_{1}$, $a_{2}$, b, c, d, and e, whose amplitudes remain unaltered even when the symmetry plane of the NV center changes. It is to be noted that Equations (5) through (9) are completely analogous in terms of the strain tensor $\varepsilon_{ij}$. We followed the representations introduced by Barson et al. [49] and implemented them in the theoretical framework developed by Barfuss and Udvarhelyi et al. [27,46]. These parameters can be expressed explicitly in term of six independent spin–stress coupling parameters, i.e., $g_{43}$, $g_{41}$, $g_{25}$, $g_{26}$, $g_{15}$, and $g_{16}$, which are reported in Table 2.

The contribution of $N_{i}$ is negligible compared to that of $M_{i}$ if the perturbation due to the applied or spin–stress-induced symmetry-breaking field is omitted. The contribution of $H_{\sigma 1}$ is much smaller than that of the zero-field splitting parameter $D_{0}$. The total Hamiltonian under stress and magnetic field can be written as
(11)$$H=D_{0}S_{z}^{2}+M_{z}S_{z}^{2}+N_{x}\left\{S_{x},S_{z}\right\}+N_{y}\left\{S_{y},S_{z}\right\}+M_{x}\left(S_{y}^{2}-S_{x}^{2}\right)+M_{y}\left\{S_{x},S_{y}\right\}+\gamma_{e}B\cdot S$$

The operators $\left\{S=S_{x},\;S_{y},\;S_{z}\right\}$ for a spin-one system are 3×3 matrices. On this basis, the eigenvalues of Equation (11) should provide the level shifting and energy splitting under an applied stress and magnetic field. We neglected the term $\left\{{N_{x},N}_{y},N_{z}\right\}$ and solved the eigenvalues of the Hamiltonian analytically. The first two eigenvalues provide the frequencies per GPa
(12)$$\omega_{\left.|\pm 1\right\rangle }={D_{o}+M}_{z}\pm \sqrt{{\left(\gamma_{e}B\right)}^{2}+{M_{x}}^{2}+{M_{y}}^{2}}$$

The second term in Equation (12), $M_{z}$ (a shift in the level (δE)), arises due to the stress interacting with the z-component of the spin. This stress clearly induces a constant level shift on the zero-field splitting parameter $D_{o}$. The third term is due to the distortion in the trigonal symmetry and introduces further splitting on the level in the $\left.|\pm 1\right\rangle$ spin manifold. In fact, the effective Hamiltonian for Equation (11) has the same form as that for the two-dimensional state of the stress tensor.

To understand the nature of the shift and the splitting, we considered a uniaxial stress that was applied along the direction p. The stress tensor $\sigma_{IJ}$, where $I,J=\left\{X,Y,Z\right\}$ are the directions representing the crystal axes, can be written as $\sigma_{IJ}=cos(p,I)cos(p,J)$ for unit stress. For the stress along the $\left[100\right]$ direction, $\sigma_{XX}=1$; the remaining diagonal elements are $\sigma_{YY}=\sigma_{ZZ}=0$, and the off-diagonal elements are $\sigma_{XY}=\sigma_{YZ}=\sigma_{ZX}=0$. Similarly, for stress along the $\left[110\right]$ direction, we have $\sigma_{XX}=\sigma_{YY}=\sigma_{XY}=1/2$ and $\sigma_{XZ}=\sigma_{YZ}=\sigma_{ZZ}=0$, and for stress along the $\left[111\right]$ direction, $\sigma_{XX}=\sigma_{YY}=\sigma_{ZZ}=\sigma_{XY}=\sigma_{YZ}=\sigma_{ZX}=1/3$. For more complex directions such as the $\left[120\right]$ one, the components of the stress tensors are $\sigma_{XX}=1/5$, $\sigma_{YY}=4/5$, $\sigma_{XY}=2/5$, and $\sigma_{XZ}=\sigma_{YZ}=\sigma_{ZZ}=0$. Introducing these values for $\left\{{M_{x},M}_{y},M_{z}\right\}$ in Equations (5)–(7), we obtained the following eigenvalues (energy level shift/split per GPa) for the $\left[100\right]$, $\left[110\right]$, and $\left[111\right]$ directions: (13)$${∆\omega }_{\left.|\pm 1\right\rangle }^{\left[100\right]}=a_{1}\pm 2b$$
(14)$${∆\omega }_{\left.|\pm 1\right\rangle }^{\left[110\right]}=a_{1}+a_{2}\pm (b-c)$$
(15)$${∆\omega }_{\left.|\pm 1\right\rangle }^{\left[111\right]}=a_{1}+{2a}_{2}$$

Equations (13)–(15) completely characterize the stress–spin interactions at a zero applied magnetic field. For the $p∥\left[100\right]$ and $\left[110\right]$ directions, there was splitting, as indicated by the signs in Equations (14) and (15). The values of this splitting were different for stress along different directions. If $p∥\left[111\right]$, the shift was a constant with a value of $a_{1}+{2a}_{2}$. In the above derivation, the NV axis was assumed to be oriented along the $e_{z}=\left[111\right]$ direction, and therefore only the z component of the spin vector S coupled to the applied stress. It was found that for p∥[100], the shift was the same for each sub-ensemble of the NV centers that were oriented in all four directions. For p∥[110], NV centers with $e_{z}\in \{111,\;\bar{1}\bar{1}1\}$ had splitting $a_{1}+a_{2}\pm (b-c)$, and NV centers with $e_{z}\in \{\bar{1}1\bar{1},1\bar{1}\bar{1}\}$ had splitting $a_{1}-a_{2}\pm (b-c)$, whereas for p∥[111], the splitting was $a_{1}+{2a}_{2}$ and $a_{1}-2a_{2}/3\pm 4c/3$, respectively, for $e_{z}\in \left\{111,\;\bar{1}\bar{1}1\right\}$ and $e_{z}\in \{\bar{1}1\bar{1},1\bar{1}\bar{1}\}$. The splitting in each case under applied uniaxial stress is provided in Table 3. With the experimental observation provided for the orientations by Doherty et al. [41], we summarize that the NV sub-ensemble orientations with respect to p were $\theta^{{}^{\circ}}=0$ for $e_{z}\in \left\{111,\;\bar{1}\bar{1}1\right\}$ and ${\theta =70}^{{}^{\circ}}$ for $e_{z}\in \left\{\bar{1}1\bar{1},1\bar{1}\bar{1}\right\}$ with $p∥\left[111\right]$ and ${\theta =36}^{{}^{\circ}}$ for $e_{z}\in \left\{111,\;\bar{1}\bar{1}1\right\}$ and ${\theta =90}^{{}^{\circ}}$ for $e_{z}\in \left\{\bar{1}1\bar{1},1\bar{1}\bar{1}\right\}$ with $p∥\left[110\right].$ For the sub-ensembles with $e_{z}\in \left\{111,\;\bar{1}\bar{1}1,\;\bar{1}1\bar{1},1\bar{1}\bar{1}\right\}$, ${\theta =54}^{{}^{\circ}}$ with $p∥\left[100\right].$ The angle of orientation affects the photon absorption rates depending on photon polarization.

In Reference [46], Udvarhelyi calculated the values of the stress susceptibility parameters d and e as −0.12 ± 0.01 and 0.66 ± 0.01 MHz/GPa, respectively, using DFT; no experimentally observed values are available yet for these parameters. They also reported the values of other parameters, i.e., $a_{1}=-2.66\pm 0.07$, $a_{2}=2.51\pm 0.06$, b=1.94±0.02, and c=−2.83±0.03 MHz/GPa, which are consistent with the experimentally measured values of 4.4±0.2, −3.7±0.2, −2.3±0.3, and −3.5±0.3 MHz/GPa, respectively [49]. The DFT calculation of these parameters was based on the method developed for the zero-field splitting of spin–spin interactions by Bodrog and Gali [66] within the projected augmented approximation [53] framework. The splitting depends on the orientation of the photon polarization vector with respect to the direction of the applied uniaxial stress. For the π polarization, where the NV axis is along the photon’s polarization vector, and the NV dipole is oriented along the $\left[111\right]$ direction, the energy shift and the splitting {δE, ∆E} of the ±1 spin manifold can be calculated from Table 1 as $\left\{3,\;0\right\}\;MHz/GPa$ for $p∥\left[111\right]$, $\left\{0.7,\;\pm 5.8\right\}\;MHz/GPa$ for $p∥\left[110\right]$, and $\left\{4.4,\;\pm 4.6\right\}\;MHz/GPa$ for $p∥\left[100\right]$. The parameters {δE, ∆E} for the NV sub-ensemble oriented along the [$1\bar{1}\bar{1}$] direction are calculated as $\left\{6.86,\;\pm 4.66\right\},$ $\left\{0.7,\;\pm 5.8\right\}$, and $\left\{4.4,\;\pm 4.6\right\}$ MHz/GPa, respectively, for $p∥\left[111\right]$, $p∥\left[110\right]$, and $p∥\left[100\right]$. It is interesting to note that with $p∥\left[111\right]$, there is no splitting of the energy level for $e_{z}\in \left\{111,\;\bar{1}\bar{1}1\right\}$ NV orientations.

Figure 6 shows the shift and the splitting of the ±1 spin manifold for the NV sub-ensemble oriented along the [111] direction with applied stress along the $\left[100\right]$ direction. Both the split and the shift increased linearly as a function of stress. 

## 4. Discussions

For the applied stress with $p∥\left[111\right]$ to the NV centers’ orientation $e_{z}\in \{111\}$, there was no splitting, and only shifting was observed. If a magnetic field was present, there was splitting with strength $\pm \gamma_{e}B$. The magnitude of the magnetic field strength could easily surpass the effect of stress. For example, a magnetic field strength of 10 mT provided a splitting of ~280 MHz, which is equivalent to applying a stress of ~60 GPa. To minimize the effect of the magnetic field in stress-sensing experiments, we suggest initially benchmarking the splitting only due to the magnetic field along the z-axis to reduce unwanted cross-sensitivity. 

The sensitivity ${(\eta }_{gs})$ of a stress-sensing device due to the manipulation of the ground state energy of a NV center is determined mainly by two factors: the split or shift of the spin levels with respect to the applied stress and the spin relaxation time in the ground state spin manifold [41]. The stress sensitivity is defined as $\eta_{gs}={\left(2\pi C\left(dD/dP\right)\sqrt{T_{2}^{*}}\right)}^{-1}$, where C is the photon collection efficiency, $\frac{dD}{dP}$ is the change in zero-field splitting with respect to the applied stress, and $T_{2}^{*}$ is the sub-ensemble spin dephasing time [3,67]. The collection efficiency could be significantly enhanced using spoof plasmonic waveguides [35,68,69]. A typical value at room temperature for a low concentration of NV centers is C~0.05 but could be enhanced up to C~0.5 by coupling the single spin rotation signal with the resonances of the spoof plasmonic waveguide. Using Table 3, for the sub-ensemble with $e_{z}\in 111$ and with a direction of the applied stress $p∥\left[111\right]$, a level shift was obtained as dD/dP=3 MHz/GPa. Using a spin relaxometry technique, in our earlier work we measured $T_{2}^{*}\approx 10\;\mu s$ for a NV center ensemble with nanodiamonds of ~70 nm average size distribution at room temperature [35]. Using these values, we estimated the sensitivity $\eta_{gs}\approx 0.32\;MPa/\sqrt{Hz}$. This is comparable to the pressure sensitivity of $0.6\;Mpa/\sqrt{Hz}$ obtained in an experiment by Doherty et al. [41]. In that experiment, a hydrostatic pressure of up to 60 GPa was applied in a diamond anvil cell (DAC) using single-crystal chemical vapor deposition (CVD)-grown diamond with a nitrogen content of <1 ppm. Experimentally, reduced ODMR contrast and count rate from a particular NV center could significantly impact the sensitivity. In addition, the spin dephasing time that was used in the current calculation was about 10 times longer than that in Doherty’s experiment [43]. This factor alone enhanced the sensitivity by $1/\sqrt{T_{2}^{*}}$ in our calculation. 

To compare the quantum sensing capability in this work with sensitivities obtained using traditional optical based sensors, the band edge shift is a reasonable metric to consider under applied stress. The traditional optical sensing limit that relies on the shift of the conduction or valence band edges could be estimated by taking our band edge shift reported in Table 1. For 26.61 GPa, the conduction band edge shift was nearly 60 meV. This is equivalent to a wavelength shift ∆λ=32 nm and an energy shift per unit pressure $\frac{dE}{dP}=$ 5.88 × 10^5^ MHz/GPa. The DFT results for band gap estimation are very rough estimates due to inaccurately accounting for self-interaction. However, the overall trend and energetics of the shifts calculated by DFT are expected to provide insights on the nature of the shifts and to yield a close estimation of the sensitivity. The value of $\frac{dE}{dP}$ calculated for the band shift using DFT was about four orders of magnitude higher than the energy shift per unit pressure in the spin manifold. This shows the superiority of the stress sensitivity that could be achieved by manipulating the ground-state spin levels in a quantum sensor compared to that of traditional optical sensors based on band edge shifting. 

Finally, based on our results, we propose the stress sensor “quantum manometer” and discuss its usefulness to detect parameters such as subsurface seismic vibrations. Subsurface pressure evolution is a critical observable quantity in geological studies used for understanding the subsurface structure, conducting oil and gas exploration, and identifying seismic vibrations [70]. Hydraulic fracturing and gas injection may lead to seismic vibrations and a series of aftershocks. Trapped high-pressure CO_2_ (pressure over 10 MPa) in the deep subsurface region can lead to series of co-seismic pulses [71]. In Reference [72], volumetric strain due to effective stress changes in the injected fluid (CO_2_) in crystalline rock was sensed by changes in the seismic velocity under applied stress up to 13–14 MPa. Using a NV center in diamond, this pressure limit can be manifested in a splitting of over 50 MHz. This is above the sensing limit of the proposed quantum manometer based on NV centers in diamond, and therefore, such changes in stress under high pressure can be easily detected. High pressure can be a controlled parameter that is tuned systematically in order to drive the systems under investigation to a desired physical state in materials science fields. One such example is metallic hydrogen, which is expected to be achievable under ultra-high pressure [73]. Ultra-high pressure detection can be incredibly challenging using normal pressure-sensing devices at the nanoscale. Surface plasmon resonances and glass transitions are other examples where a systematic control of pressure in extreme conditions must be conducted. A most striking example of this could be high-temperature superconductivity with transition temperatures (Tc) between 250 and 260 K, where pressure detection and monitoring are essential up to 200 GPa [63,74]. In all of the above examples, our proposed model device could find an application for sensing pressure under extreme conditions.

## 5. Conclusions

NV centers in diamond are a promising quantum material due to their optical properties that allow for the achievement of sensitivity levels that are several orders of magnitude lower than those of their classical counterparts. By combining the DFT results, we calculated the energy level shift and splitting of the spin states at the ground energy level. The results of the density of states, bandstructures, and strain applications were presented and showed up to a 60 meV conduction band edge shift over 25 GPa of pressure applied along the longitudinal and transverse crystallographic directions. By engineering a low-energy Hamiltonian for a ±1 spin manifold, we calculated the zero-energy shift per unit of applied stress and obtained a sensitivity of $0.32\;MPa/\sqrt{Hz}$ for a NV sub-ensemble aligning to the [111] direction under pressure applied along the same direction. The energy shift per unit pressure of the quantum sensor was found to surpass that of traditional optical sensors (which operate based on band edge shifting) by several orders of magnitude. Our results are expected to be useful for designing pressure sensors that can operate under high-pressure environmental conditions. 

## Figures and Tables

**Figure 1 nanomaterials-14-00675-f001:**
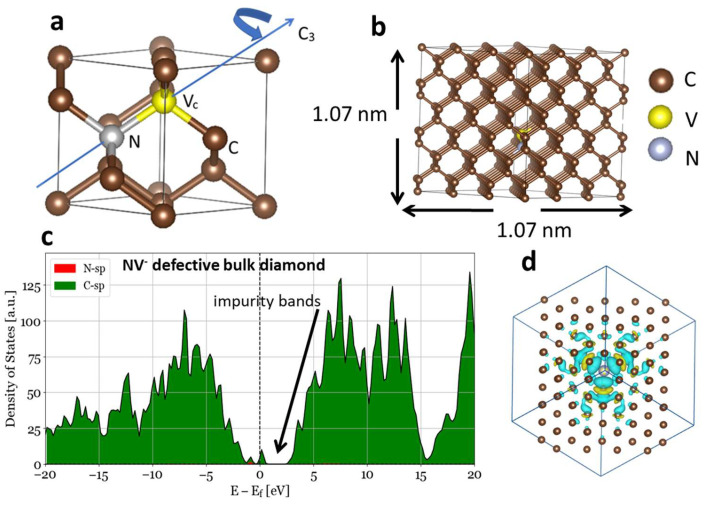
A diamond structure with a NV center. Shown are (**a**) the unit cell and (**b**) a 3 × 3 × 3 supercell structure with dimensions of 1.07 nm. The DOS for NV^−^ defective diamond. (**c**) The impurity states can be seen around the Fermi level with some contributions from C. (**d**). The charge distribution around the N impurity and C vacancy centered around the supercell with charge q=−1. The color map in (**d**) indicates positive (yellow) and negative (blue) charge surfaces.

**Figure 2 nanomaterials-14-00675-f002:**
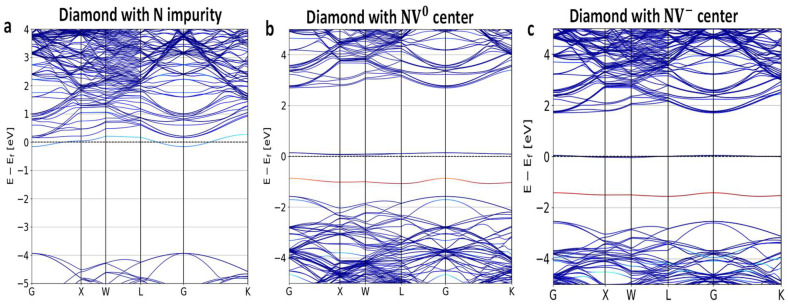
Diamond bandstructures calculated using the PAW-PBE potential (**a**) with a N impurity, (**b**) NV^0^, and (**c**) with NV^−^ centers in a 3 × 3 × 3 supercell for a NV center oriented along the [111] direction. As a NV center was introduced, additional defect bands arose with a band gap of nearly 1.5 eV in this calculation, allowing for strong electron-hole polarizations under green laser illumination in both neutral and negative NV defects. Dark blue denotes the C states, whereas red and light blue indicate impurity states (zero energy is placed at the Fermi level).

**Figure 3 nanomaterials-14-00675-f003:**
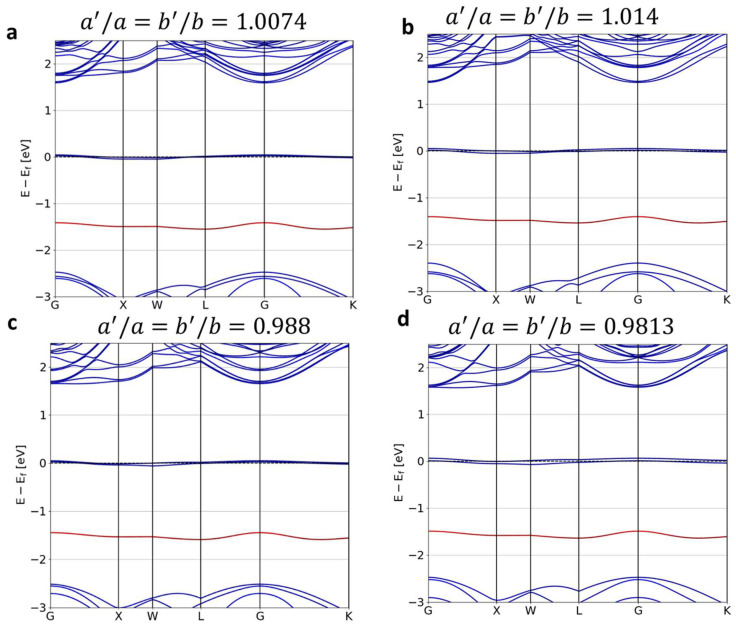
NV-center diamond band structures for changes in the transverse lattice parameter ratio $\frac{a^{\prime }}{a}=\frac{b^{\prime }}{b}$ by (**a**) 1.0074, (**b**) 1.014, (**c**) 0.988 and (**d**) 0.9813. The blue curves represent the C bands. whereas the red curves represent the impurity band.

**Figure 4 nanomaterials-14-00675-f004:**
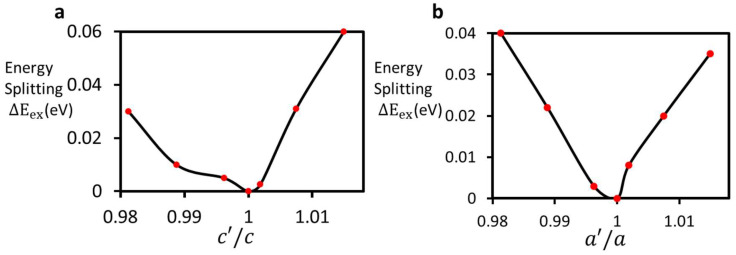
Energy splitting of the excited ^3^E level under a variation in the ratios of the longitudinal (**a**) and transverse (**b**) lattice parameters. The transverse lattice parameters a and b were changed simultaneously in (**b**) up to 2%. Only the change in $a^{\prime }/a$ is shown, due to the cubic symmetry of the supercell.

**Figure 5 nanomaterials-14-00675-f005:**
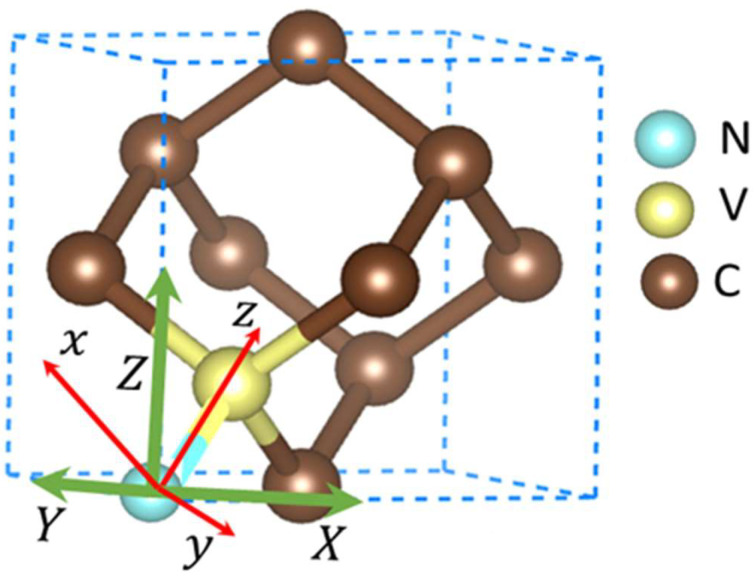
NV center in a diamond unit cell showing coordinate systems with respect to NV axes (red) and crystallographic axes (green) [63].

**Figure 6 nanomaterials-14-00675-f006:**
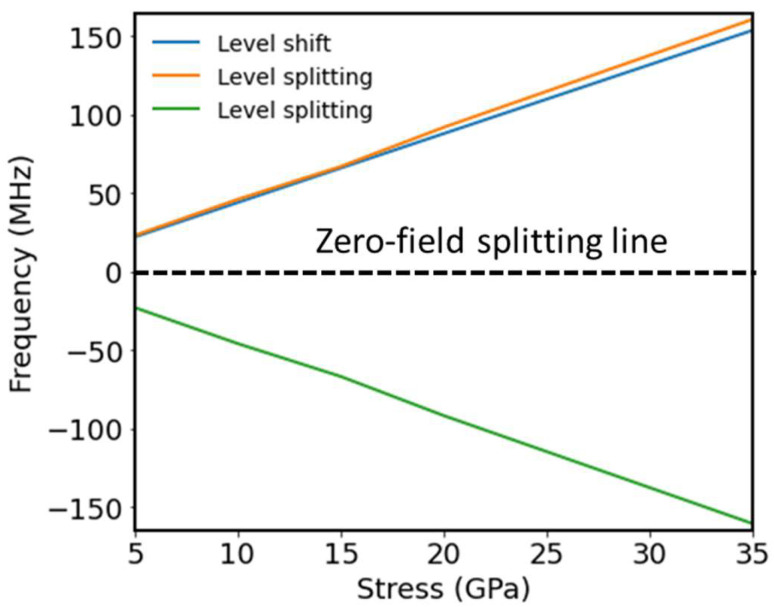
Energy level shift (blue) and splitting (red and green) of the ±1 spin manifold as a function of the applied stress along the $p∥\left[100\right]$ direction for the NV centers with orientation $e_{z}\in \{111\}$ in a zero magnetic field. Zero-field splitting is aligned with the zero energy.

**Table 1 nanomaterials-14-00675-t001:** Splitting and changes in the band gap magnitude under variation in c (Figure 4a) and a and b (Figure 4b) in a 3×3×3 supercell.

Change in Lattice Parameter c (Å)	Splitting of Excited State $∆E_{ex}$ (meV)	Band Gap Changes (meV)	Change in Lattice Parameters a and b (Å)	Splitting of Excited State $∆E_{ex}$ (meV)	Band Gap Changes (meV)
10.87	30	50	10.87	35	55
10.79	10	30	10.79	20	40
10.71	0	0.0	10.71	0	0.0
10.67	10	30	10.67	3	15
10.59	31	25	10.59	22	25
10.51	60	30	10.51	40	35

**Table 2 nanomaterials-14-00675-t002:** Stress susceptibility parameters.

$a_{1}=\left(2g_{41}+g_{43}\right)/3$	$a_{2}=\left(-g_{41}+g_{43}\right)/3$	$b=\left(-g_{15}+\sqrt{2}g_{16}\right)/3$
$c=\left(-2g_{15}-{\sqrt{2}g}_{16}\right)/3$	$d=\left(-g_{25}+{\sqrt{2}g}_{26}\right)/12$	$d=\left(-{2g}_{25}-{\sqrt{2}g}_{26}\right)/12$

**Table 3 nanomaterials-14-00675-t003:** Energy splitting for four different NV center orientations under applied uniaxial pressure along three different directions.

Applied-Stress Direction	NV Sub-Ensemble Direction	Shift/Splitting per Unit Pressure
p∥[100]	$e_{z}\in \{111,\;\bar{1}\bar{1}1,\;\bar{1}1\bar{1},1\bar{1}\bar{1}\}$	$a_{1}\pm 2b$
p∥[110]	$e_{z}\in \{111,\;\bar{1}\bar{1}1\}$	$a_{1}+a_{2}\pm (b-c)$
	$e_{z}\in \{\bar{1}1\bar{1},1\bar{1}\bar{1}\}$	$a_{1}-a_{2}\pm (b-c)$
p∥[111]	$e_{z}\in \{111,\;\bar{1}\bar{1}1\}$	$a_{1}+{2a}_{2}$
	$e_{z}\in \{\bar{1}1\bar{1},1\bar{1}\bar{1}\}$	$a_{1}-2a_{2}/3\pm 4c/3$

## Data Availability

Data are contained within the article.

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
