# Peer review of "Sensing at the Nanoscale Using Nitrogen-Vacancy Centers in Diamond: A Model for a Quantum Pressure Sensor"

_nanomaterials, 2024, doi:10.3390/nano14080675_

Round 1

Reviewer 1 Report

Comments and Suggestions for Authors

The work "Stress Sensing at Nanoscale Using Nitrogen Vacancy Centers in 2 Diamond: A Model for Quantum Manometer " presents a deep theoretical insights into the nitrogen vacancy centers in diamond. It is shown that by strain engineering a high sensitivity of 0.32 MPa/√Hz is obtained. Therefore, the nitrogen vacancy centers in diamond are a promising quantum material. The work is well written and organized. The conclusions are supported by the results. The topic is of interest of readers of the Nanomaterils journal, however, it may be more related to applied quantum mechanics. The following comments should be addressed before I can recommend the work for publication:

The title may need to be reconsidered. A part "A Model for Quantum Manometer " never repets in the text. Therefore, it is better to rephrase the title or add discussion on a "Quantum Manometer ".

It suggested to discuss the concentration of nitrogen vacancy centers based on the struture size used.

Please add more information on the methogology for the calculation of the charge distribution. In addition, in Figure 1d the color markers for the charge density are needed.

It is said that Monkhorst-pack grid of 9×9×9 was used for the unit cell. What k-mesh grid was used for the 3x3x3 supercell?

Reviewer 2 Report

Comments and Suggestions for Authors

Manuscript ID: nanomaterials-2930439

Comments and Suggestions:

This research article presents a novel and promising approach for stress sensing at the nanoscale using nitrogen vacancy (NV) centers in diamond, proposing a model for a quantum manometer. The integration of first-principles density functional theory (DFT) simulations and theoretical modeling to investigate the effect of strain on NV centers in diamond demonstrates the potential for unprecedented levels of pressure sensing. The article discusses the development of a low energy Hamiltonian to predict the sensitivity of NV centers to pressure variations, showing superiority over traditional optical sensing techniques. Additionally, the proposed quantum manometer offers potential applications in several fields including earth's subsurface scanning and resource recovery.

Suggestions for Major Revision:

1. Provide a clearer delineation of the theoretical framework used, particularly in terms of the mathematical models employed for DFT simulations and Hamiltonian development. This will help readers, especially those less familiar with quantum mechanics, to better understand the theoretical basis of the research.

Enhanced Methodology Description:

2. Expand the section on computational methods to include more detailed explanations of key parameters, such as the rationale behind the choice of specific lattice parameters for stress application and the computational protocols for bandstructure calculations. This will improve reproducibility and facilitate comparisons with other studies.

3. Present a more comprehensive overview of the obtained results, possibly including additional figures or tables to illustrate key findings. This could involve showcasing the density of states and bandstructures under different stress conditions more explicitly, providing a clearer visualization of the observed shifts and splits in energy levels.

4. Address the assumptions and limitations inherent in the theoretical models and computational simulations employed. Discuss any simplifications or approximations made and their potential impact on the accuracy of the predictions. This will add depth to the discussion and help readers assess the robustness of the conclusions.

5. Where possible, compare the theoretical predictions with experimental data or other theoretical models from the literature. This will validate the proposed model and provide a basis for assessing its predictive power. Additionally, discussing discrepancies between theory and experiment can offer insights into areas for future refinement.

6. Expand the discussion on the potential applications of the proposed quantum manometer beyond the specific examples mentioned in the abstract. Consider addressing how this technology could impact other fields beyond earth sciences and resource exploration. Additionally, suggest avenues for future research aimed at optimizing the sensitivity and versatility of NV center-based pressure sensors.

Recommendation for Review Decision:

Given the substantial theoretical and computational work presented in this article, along with the potential impact on sensing technology and scientific understanding, I recommend accepting this manuscript for publication in Nanomaterials after the major revisions suggested above have been implemented.

Comments on the Quality of English Language

Moderate editing of English language required.

Round 2

Reviewer 2 Report

Comments and Suggestions for Authors

Review (2nd round)

Manuscript ID: nanomaterials-2930439

Comment and Recommendation:

The author has addressed the queries raised by the reviewers point by point and has made a fairly comprehensive and thorough revision in accordance with the reviewers' suggestions. While, the revisions made still lack meticulousness and require further improvement. Therefore, I recommend that the current research paper be accepted for publication in Nanomaterials with the condition that further revisions be made to ensure its quality and coherence.

Comments on the Quality of English Language

Moderate editing of English language required.

Author Response

The authors thank the reviewer for the comment. The authors revised the manuscripts in accordance with the reviewers' comments in the 1st round and added sufficient texts in the manuscript. While the comments raised by reviewers in the 1st round greatly increased the value of the manuscript, the comment received in the 2nd round on the "meticulousness" was not clear to authors. Nevertheless, in the 2nd revision, the authors further improved the manuscript by revising the ambiguous sentences and words.  The citations were also added. In addition, the manuscript was thoroughly evaluated by our professional editors and native English researchers with expertise on the presented works in order to eliminate typos and absurd sentences.